# Health-Related Quality-of-Life Outcomes in Patients with Recessive and Dominant LGMD: A Comparative Cross-Sectional Study

**DOI:** 10.3390/muscles4030025

**Published:** 2025-07-30

**Authors:** Clara Lépée-Aragón, Irune García, Alicia Aurora Rodríguez, Corrado Angelini, Oscar Martínez

**Affiliations:** 1Neuro-e-Motion Research Team, Faculty of Health Sciences, University of Deusto, Av. Universidades 24, 48007 Bilbao, Spain; c.lepee@deusto.es (C.L.-A.); aliciarodriguez.b@deusto.es (A.A.R.); oscar.martinez@deusto.es (O.M.); 2Neuromuscular Laboratory, Department of Neurosciences, University of Padova, Campus Biomedico Pietro d’Abano, 35131 Padua, Italy; corrado.angelini@unipd.it

**Keywords:** LGMD, recessive, dominant, HRQoL, patient profile, clinical practice

## Abstract

Limb–girdle muscular dystrophy (LGMD) encompasses a heterogeneous group disease, genetic and phenotypically. There are more than 30 subtypes divided into two groups: autosomal dominant and recessive. LGMDs are characterised by muscle weakness; however, psychosocial factors seem to be affected too, such as HRQoL. Given the lack of literature in this respect, the present cross-sectional study aimed to create a patient profile comparing both dominant and recessive forms by analysing HRQoL through the INQoL, and sociodemographic data. The LGMD-recessive group had a worse HRQoL compared to the dominant group, specifically in the dimensions of muscle weakness (*p* = 0.007), emotion (*p* = 0.046), independence (*p* = 0.029), and body image (*p* = 0.022). In addition, in the LGMD-dominant group, 77.9% of the relational indicator was explained by age (B = 0.907, *p* = 0.012), which can be understood as a limitation in their social role due to the disease progression. In contrast, no sociodemographic variables were found to be predictive of the HRQoL of patients with recessive forms of LGMD. These results are relevant for clinical practice, as they reflect the most affected areas of HRQoL in LGMD patients, differentiating between recessive and dominant forms.

## 1. Introduction

Limb–girdle muscular disease (LGMD) is a heterogeneous group of neuromuscular disorders which are characterised by a proximal muscle weakness, principally affecting the girdle and shoulder muscles. This results in elevated creatine kinase (CK) levels and dystrophic changes on muscle biopsy [1,2]. Some subtypes of LGMD suffer from cardiac and pulmonary complications, limiting the life expectancy of some affected individuals [3,4].

Clinically, LGMDs have a very diverse genotypic and phenotypic profile with several subtypes divided in two different categories, recessive and dominant [5]. On the one hand, LGMD-D, with autosomal dominant inheritance, comprises five subtypes (LGMD-D1-D5), while LGMD-R, with an autosomal recessive pattern of inheritance, constitute 24 subtypes (LGMD-R1-R24) [6]. According to their prevalence, both forms are considered rare diseases, but recessive forms are more frequent than dominant [7]. Dominants share many clinical features with their recessive counterparts, but often have a later age of onset and milder course [8]. In the same way, the progressive course is slower in comparison to recessive ones [9]. People with recessive subtype have worse symptoms compared to the dominant subtype. Generally, serum CK is much lower in the dominant group than in the recessive group [9].

Unlike other neuromuscular diseases, LGMD is a long-lasting disease, which can present at early or late ages, and progresses with advancing age [10]. In fact, patients who are diagnosed with LGMD before the age of 30 years old have faster and worse degeneration, causing the loss of ambulation [11]. That is why an early onset is correlated with severe prognosis [12]. In addition, the comorbidities and physical limitations that accompany the disease not only make it difficult to perform basic activities of daily living, but also have an impact on the patient’s social and emotional life [4].

In general, the clinical profile of patients is more characterised by physical impairment rather than mental disability, in contrast to the other muscular dystrophy diseases [10]. Some subtypes of LGMD suffer from cardiac and pulmonary complications, limiting the life expectancy of some affected individuals [3,4]. Despite advances in research, in the absence of a definitive cure for LGMD [13], it is important that the physical, but also the psychological and social aspects of these patients are considered. Thus, the analysis of health-related quality of life (HRQoL) is essential [14].

Some of the few previous studies that have been conducted on the HRQoL of LGMD patients have focused on the physical implications of their symptoms [3,4,10,15,16]. For example, patients with LGMD may feel very tired after prolonged physical exertion, requiring rest to compensate [14]. This daily situation is frustrating for them and represents a handicap in their socialisation, fostering their tendency towards isolation. Although physical symptoms are more prominent, a recent study suggested that psychological symptoms are equally significantly important for people with LGMD (specifically, emotional distress, impaired body image and social role dissatisfaction) [3]. In this sense, analysing the psychosocial aspects of the HRQoL of patients with LGMD is crucial [16].

Nevertheless, there are hardly any studies that have analysed the combined physical and psychosocial aspects of HRQoL in LGMD patients [14], let alone compared dominant and recessive forms. This would be an important aspect given the genotypic and phenotypic variability of LGMD, which would allow interventions to address their specific needs and improve the clinical management of the patient [14]. This analysis should include factors such as illness, support, resources and life expectancy, and to place these factors in the illness evolution [4].

In response, the aim of this study is to analyse the physical and psychosocial aspects of HRQoL in a multicultural cohort of adult patients with different types of LGMD, while providing a comparison between dominant and recessive forms. This will provide an overview of the specific needs of LGMD patients.

## 2. Methods

### 2.1. Participants

A total of 19 adult LGMD patients (11 women and 8 men) with Spanish, Italian, and/or Hungarian nationality were included in this study. The participants were aged between 18 and 58 years and were recruited through different patient centres. The Spanish participants were enrolled from the Association of Limb–Girdle Muscular Dystrophy due to Sarcoglycan deficiency (Proyecto Alpha), whereas Italian and Hungarian patients came from the association “Conquistando Escalones” and a hospital in Padua.

The inclusion criteria were (a) a diagnosis of LGMD made by a neurologist; (b) being of legal age; and (c) having Spanish, English, or Italian as one of their main languages. The exclusion criteria were (a) the presence of any other diagnosis or sensory deficit that would prevent the application of the tests and (b) being illiterate.

### 2.2. Instruments

The evaluation protocol was administered in its Spanish, Italian, and English versions. This is a recognised instrument in terms of validity and reliability among adults affected by a neuromuscular disease. Although it has been recommended for use in LGMD, especially to compare aspects of HRQoL between recessive and dominant forms [4], it has not yet been widely used, as it is one of the few valid and reliable tests that measures quality of life specific to muscle diseases [17].

#### Individualized Neuromuscular Quality of Life (INQoL)

The INQoL test measures the HRQoL of adults with neuromuscular diseases developed by Vincent and cols. [17]. In this study, the INQoL’ English [17], Spanish [18], and Italian [19] versions were administered. The time estimated to pass the scale is approximately 15 min. It is a self-administered questionnaire that it consists of 45 items, divided into nine groups that measure (1) muscle weakness, (2) locking, (3) muscle pain, (4) fatigue, (5) activities, (6) independence, (7) relationships, (8) emotions, and (9) body image, which are described in Table 1. Each category has several items that are measured in a 7- or 8-point Likert scale. Furthermore, the first four groups (muscle weakness, muscle locking, pain, and tiredness) are introduced by a yes/no question, asking whether the symptoms are present or not. If the answer is negative, it is necessary to jump to the next group, while if the answer is positive, more questions are suggested. The total score is between 0 and 100; the higher the score, the worse the condition the patient is in. The Cronbach’s alpha of this instrument is 0.8, which means that it has a good internal consistency.

### 2.3. Procedure

A cross-sectional research was performed with a convenience sample due to the low incidence of the current clinic population. Patient associations and hospitals distributed the information about the project, and interested patients were then contacted by the researchers. Due to the geographical dispersion of this rare disease, the test has been administered via online questionnaire. For this purpose, a ‘Google Form’ link was sent by the researchers to each participant to fill in the test. Moreover, questions related to sociodemographic data were asked in the same link. The time estimated to complete the form was less than half an hour. In addition, the researchers were attentive to any questions patients might have when filling in the form. All the assessments were performed under similar environmental conditions. Finally, participants were recruited during the years 2024–2025.

Before its implementation, all the participants completed an informed consent form and agreed to participate voluntarily in the study. The project was approved by the Ethics Committee at the University of Deusto (Ref: ETK-62/23-24) and was conducted following the ethical principles established by the Declaration of Helsinki.

### 2.4. Data Analysis

The SPSS (Statistical Package for the Social Sciences) version 28 was used to perform the statistical analyses. Before the inferential analyses, the Shapiro–Wilk test was applied to determine the normal distribution of the variables. The descriptive data of the sample were reported in terms of means, standard deviations, frequencies, and percentages. Differences between groups (recessive vs. dominant) on categorical and continuous variables were analysed using the Chi-square and the Mann–Whitney *U* statistic, respectively. To measure effect size, the Rosenthal r coefficient was calculated, where r = 0.1 is considered small, r = 0.3 medium, and r = 0.5 large [20]. Moreover, Spearman’s Rho statistic was used to analyse the correlations between the sociodemographic variables and the different HRQoL dimensions in each of the groups. Finally, to support previous results, multiple regression analyses were conducted between the sociodemographic variables and significant HRQoL indicators. For this purpose, the scores were transformed into Z-scores. The level of significance was set at *p* < 0.05 in all analyses.

## 3. Results

### 3.1. Sociodemographic and Clinical Data of the Sample

Table 2 presents the sociodemographic and clinical data of the sample. There were no differences between the recessive and dominant forms for sex, χ^2^(1) = 0.224, *p* = 0.636, country, χ^2^(2) = 5.732, *p* = 0.057, or age (*U* = 37.500, *Z* = −0.132, *p* = 0.895).

### 3.2. Descriptive INQoL Scores in LGMD Sample

The HRQoL results analysed through the INQoL in the present sample of LGMD patients show the following mean scores: muscle weakness 73.9 ± 18.78; locking 26.31 ± 25.72; muscle pain 33.45 ± 27.67; fatigue 58.52 ± 18.99; activities 59.96 ± 19.97; independence 69.00 ± 26.93; relationship 29.40 ± 17.27; emotions 38.74 ± 23.67; and body image 49.70 ± 19.49. These results indicate that the most affected dimensions of HRQoL in LGMD are muscle weakness, fatigue, activities, and independence (Figure 1).

### 3.3. Difference Analyses of HRQoL Between Recessive and Dominant LGMD

The scores of the different dimensions of the INQoL between the recessive and dominant LGMD groups are shown in Table 3. In general, the recessive group has a worse HRQoL. Specifically, the findings show that the recessive group has a significant higher score comparing to the dominant group in the following variables: muscle weakness (*U* = 9.000, *Z* = −2.637, *p* = 0.007), independence (*U* = 14.000, *Z* = −2.202, *p* = 0.029), emotion (*U* = 16.000, *Z* = −2.020, *p* = 0.046), and body image (*U* = 13.500, *Z* = −2.246, *p* = 0.022), which means a worse HRQoL in those dimensions. Considering the effect size indicator, the magnitude of these differences is large in all cases. However, no statistically significant differences were found in the remaining HRQoL variables between the recessive and dominant LGMD groups. Figure 2 shows more graphically the distribution of each group in all dimensions analysed by INQoL, providing a profile of HRQoL in both groups.

### 3.4. Correlations Between the Sociodemographic Variables and HRQoL

When analysing the correlations obtained in the dominant LGMD group, the results showed that only age and the relationship indicator presented a significant and positive association (Rho = 0.943, *p* = 0.005) (Table 4). This means that older age correlates with greater relational difficulties. In contrast, in the recessive group, a significant and positive correlation was found between age and the activities indicator (Rho = 0.587, *p* = 0.035) (Table 5). This association reflects that older age is associated with increasing difficulties in performing basic activities of daily living. No statistically significant correlations were found among the remaining variables.

### 3.5. Multiple Regression Analyses Between the Sociodemographic Variables and HRQoL

Multiple regression analyses were used to assess the influence of different, sociodemographic variables, specifically the age, on different HRQoL scores reported by the patients with dominant and recessive LGMD forms. In the dominant group, there is an association between age and relationship (B = 0.907, *t* = 4.320, *p* = 0.012). Age, which is accompanied by the progression of the disease itself, explains 77.9% of the variance of the variable relationships. Conversely, age is not a predictable variable for the activity HRQoL indicator in patients with recessive LGMD form (B = 0.463, *t* = 1.734, *p* = 0.111).

## 4. Discussion

The present study analyses the differences in the physical and psychosocial HRQoL profile of patients with recessive and dominant forms of LGMD. To date, this comparison has mainly been performed by comparing case studies from the scientific literature [4], but not from a cross-sectional study.

In a way, the results of this study go in line with the scientific literature, showing that the most affected domains of HRQoL in patients with LGMD are muscle weakness, fatigue, activities, and independence. The same results have been corroborated in the study by Angelini and Rodríguez [4]. What is more, autosomal recessive LGMD subtypes are more prevalent in the present sample and these patients present a worse symptomatology than dominant forms [7]. In the current research, the most affected aspects of HRQoL among differences were muscle weakness, independence, emotion, and body image. Peric et al. [10] have shown worse scores in muscle weakness and independence, whereas they obtained better scores in relationship and emotion. However, social limitations such as emotional distress, impaired body image, and social role dissatisfaction have also been reported in other studies [3,21]. Moreover, this study shows that these social aspects increase with age, as the older the person is, the more the disease progresses, and therefore, so do its symptoms. In addition, the present cross-sectional research may shed some light providing methodological and empirical support on the different cases reported in the study by Angelini and Rodríguez [4], which was an aspect that was absent in the scientific literature.

LGMD is a progressive and degenerative disease [22], which means that symptoms of muscle weakness experienced by patients with LGMD often worsen over time [23]. In this way, age is an important factor to consider when evaluating adults with LGMD [22], so it could be understood as an indicator of the disease progression. According to the present results in different domains of HRQoL evaluated, in the case of dominant forms, there is a correlation between age and the impact on social relationship. This means that as the years go by and the disease progresses, the difficulty in dealing with social relationships will increase. There is no literature in this respect concerning LGMD patients, but these results can be compared with cases of other neuromuscular diseases, in which social relationship is also a HRQoL dimension that is seriously affected [17]. Age could be a predictor variable, as it is associated with certain social and relational roles that may be affected due to the disease progression. In the recessive forms, although there is a correlation between age and the activity indicator, the ability to perform daily activities, the regression analyses showed no significant results between these variables. There is no literature in this respect concerning LGMD patients.

Although the physical differences between the recessive and dominant forms of LGMD are already known, which is confirmed by the fact that the recessive group shows a more affected level of performance in the indicators of muscle weakness and independence, these differences also extend to psychosocial parameters such as well-being and self-image satisfaction, with patients in the recessive LGMD group also showing greater impairment. Furthermore, factors such as age, which can be understood as an indicator referring to the disease progression, only interfere with psychosocial aspects, like social interaction. Interestingly, this was only found in the group of patients with dominant forms of LGMD. A possible explanation may be attributed to the coping strategies adopted at an earlier age in the recessive forms. Despite a worse physical symptomatology due to an early-age onset, the coping strategies these patients may learn lead to an adaptive psychosocial functioning [16]. Psychological aspects such as self-efficacy [24], understood as an individual’s confidence in their ability to overcome problems, may provide greater insight into HRQoL than physical impairment alone. Furthermore, self-efficacy has been shown to be positively associated with HRQoL in different clinical conditions [10]. Similarly, previous studies have pointed out different psychosocial factors that may be related to HRQoL such as adaptive coping, psychological flexibility, hope, resilience, and illness perceptions [23]. In this regard, it might be interesting to analyse the effect of these psychological variables in order to better understand and support the clinical management and HRQoL in patients with LGMD.

Likewise, a loss of muscle strength limits the ability to walk and perform functional tasks, which are both facets that contribute to HRQoL. These consequences can affect both the patient’s social and work activities [10]. This evidence on the psychological aspects may justify the multidisciplinary approach to treatment with counselling and social work, as they could improve social role limitations and dissatisfaction. These data remind clinicians of the importance of addressing the social and emotional difficulties that arise with progressive loss of function [21]. In addition, having knowledge of patients’ quality of life in their specific situations would help clinicians to anticipate the effect that treatment may have on them, and to intervene in their psychosocial adjustment and to improve their coping skills. Thus, the distinctive aspect of this research is to examine HRQoL, considering different domains, and to determine how it affects people with different diagnoses of LGMD [25]. In this way, clinicians may use evidence to improve early diagnosis, personalise treatment plans and enhance patient-monitoring protocols [14,16].

### Limitations and Future Directions

Despite the findings of the present study, there are some limitations to take into account. First of all, LGMD is considered a rare disease with a low prevalence that is appreciated in the reduced sample of the study. Furthermore, the proportion between recessive and dominant forms was not equal. Therefore, statistically significant findings should be interpreted considering this limitation. Secondly, although the INQoL is an instrument adapted to different languages and widely used for physical and psychosocial HRQoL in the neuromuscular population, no more instruments have been included in the protocol. In addition, this instrument is not adapted to children, which limited the sample, excluding forms that have their onset in paediatric age. A third relevant limitation is the geographical dispersion between participants, although this reflects the frequency of cases with different forms of LGMD [26]. To some extent, the participation of more patients from other countries provides greater evidence of the impact of the disease in their HRQoL and the needs required. In any case, it is important not to forget that cultural and sociodemographic discrepancies can potentially be determinants of patients’ HRQoL [27]. Finally, the assessments were conducted by different psychologists depending on the language of the participants.

For future lines of research, it is essential to consider more studies in which a larger and more multicultural sample is included. In addition, longitudinal studies should be conducted by analysing the progression of the disease in order to detect differences that will contribute to a readjustment on their specific needs in HRQoL. Also, it is essential to include more HRQoL instruments and other potentially related variables (physical, psychological, and social) that can explain differences that current sociodemographic data are not able to.

## 5. Conclusions

This research shows an urgent necessity to investigate more on how to improve the HRQoL of people with LGMD, more specifically in the psychosocial domain. Even though recessive types have a worse prognosis, patients with dominant types are still presenting needs that should be addressed. Due to the present study, it has been possible to highlight some relevant areas to the HRQoL of patients with LGMD, which will provide information to lay the foundations for psychological intervention according to their needs.

Considering that the LGMD group involves a wide genetic and phenotypic range, it is essential to consider the needs according to the classification as much as possible. In this sense, this will allow for developing treatments, with their respective follow-up, on the specific needs presented in patients with recessive LGMD and in patients with dominant LGMD, attending to their particularities from a person-centred approach.

## Figures and Tables

**Figure 1 muscles-04-00025-f001:**
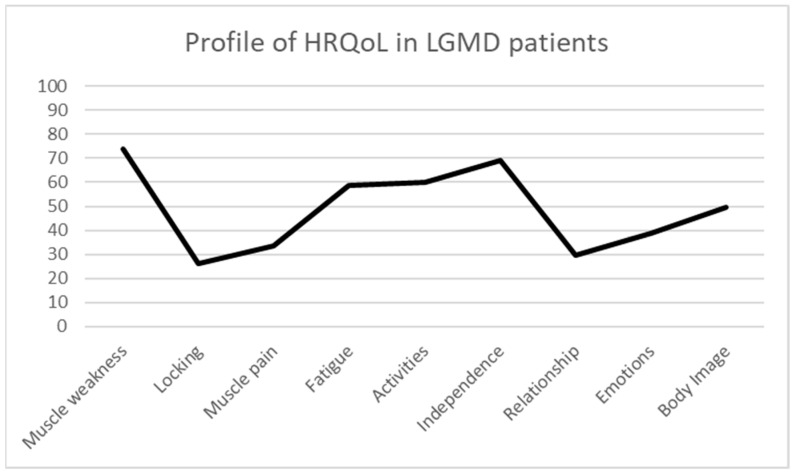
Profile of HRQoL in LGMD patients.

**Figure 2 muscles-04-00025-f002:**
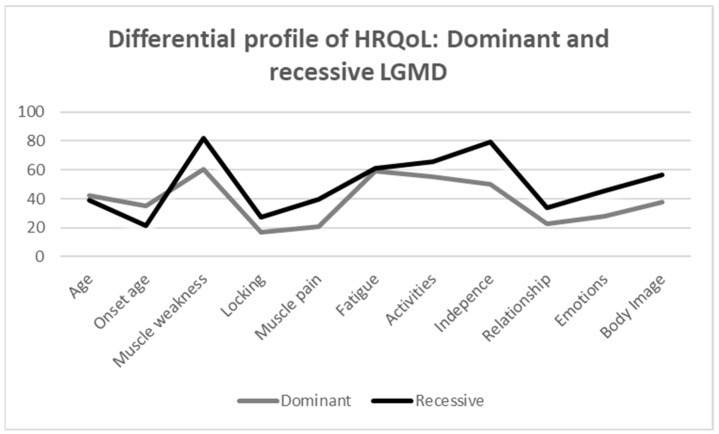
Differential profile of HRQoL: dominant and recessive LGMD.

**Table 1 muscles-04-00025-t001:** Description of the INQoL test.

INQoL Subscale	Description of Subscale
Muscle weakness	Measures the severity and impact of muscle weakness
Locking	Assesses the impact of muscle stiffness or “locking”
Muscle pain	Evaluates the level and impact of muscle-related pain
Fatigue	Measures the severity and impact of muscle fatigue
Activities	Measures the impact of the disease on the ability to perform daily activities like bathing, dressing, and housework
Independence	Assesses how much the disease affects the patient’s ability to be independent in daily life
Relationships	Explores the impact of the disease on social interactions and relationships
Emotions	Investigates the impact of the disease on mood and emotional well-being
Body image	Assesses the impact of the disease on how patients perceive their body image

**Table 2 muscles-04-00025-t002:** Sociodemographic and clinical characteristics of dominant and recessive group.

Clinical Variables	Recessives (n = 13)	Dominants (n = 6)
	N%	N%
Male/Female	5 (38.5%)/8 (61.5%)	3 (50%)/3 (50%)
LGMD type		
LGMD-D2	-	6 (31.6%)
LGMD-R2	1 (5.3%)	-
LGMD-R3	6 (31.6%)	-
LGMD-R4	2 (10.5%)	-
LGMD-R5	4 (21.1%)	-
Country		
Spain	10 (76.9%)	2 (33.3%)
Italy	3 (23.1%)	2 (33.3%)
Hungary	-	2 (33.3%)
	Mean ± SD	Mean ± SD
Age (yrs)	38.92 ± 12.952	42.50 ± 20.372
Onset age (yrs)	21.15 ± 10.629	35.25 ± 18.786

*Note:* n = participant number; *SD* = standard deviation.

**Table 3 muscles-04-00025-t003:** Mean scores and differences in clinical variables between adults with recessive and dominant LGMD.

Clinical Variables	Recessives (n = 13)	Dominant (n = 6)				
	Mean ± SD	Mean ± SD	*U*	*Z*	*p*	*r*
Muscle weakness	17 ± 1	10.83 ± 2.32	9	−2.637	0.008 **	0.605
Locking	4.67 ± 8.08	4.66 ± 5.5	35.5	−0.314	0.754	−
Muscle pain	6 ± 5.57	3.66 ± 3.72	23.5	−1.374	0.169	−
Fatigue	13.67 ± 1.15	10.16 ± 3.31	29.5	−0.837	0.403	−
Activities	20 ± 4	14.16 ± 6.67	18	−1.843	0.065	−
Independence	16 ± 0	8.33 ± 5.92	14	−2.202	0.028 *	0.505
Relationship	18.33 ± 12.58	11.5 ± 9.39	19	−1.756	0.079	−
Emotions	16.67 ± 11.24	8.83 ± 6.08	16	−2.020	0.043 *	0.463
Body image	8 ± 2	6.33 ± 3.14	13.5	−2.246	0.025 *	0.515

*Note:* n = participant number; *SD* = standard deviation; *U* = *U* Mann–Whitney; *Z* = z punctuations; *p* = level of probability; * *p* ≤ 0.05; ** *p* ≤ 0.001; *r* = effect size.

**Table 4 muscles-04-00025-t004:** Correlation between age, onset age, and clinical variables in dominant group.

	Age	Onset Age	*MW*	*L*	*MP*	*F*	*A*	*I*	*R*	*E*	*B*
Age	1.000										
Onset age	1.000 **	1.000									
Muscle weakness	0.771	0.800	1.000								
Locking	0.091	0.258	−0.334	1.000							
Muscle pain	0.232	−0.105	0.145	0.400	1.000						
Fatigue	0.486	0.400	0.429	−0.516	0.232	1.000					
Activities	0.143	0.000	0.657	−0.880 *	−0.058	0.543	1.000				
Independence	0.319	0.211	0.812 *	−0.708	−0.044	0.464	0.928 **	1.000			
Relationship	0.943 **	0.800	0.886 *	−0.152	0.290	0.657	0.429	0.580	1.000		
Emotions	0.319	0.400	0.551	−0.739	−0.221	0.754	0.754	0.794	0.551	1.000	
Body image	0.617	0.632	0.926 **	−0.393	−0.031	0.370	0.679	0.892 *	0.772	0.705	1.000

*Note:* n = participant number; *SD* = standard deviation; *U* = *U* Mann–Whitney; *Z* = z punctuations; *p* = level of probability; * *p* ≤ 0.05; ** *p* ≤ 0.001; *r* = effect size.

**Table 5 muscles-04-00025-t005:** Correlation between age, onset age, and clinical variables in recessive group.

	Age	Onset Age	*MW*	*L*	*MP*	*F*	*A*	*I*	*R*	*E*	*B*
Age	1.000										
Onset age	0.689 **	1.000									
Muscle weakness	0.111	0.044	1.000								
Locking	−0.207	−0.388	−0.327	1.000							
Muscle pain	0.150	0.046	−0.088	0.168	1.000						
Fatigue	0.130	−0.010	−0.089	−0.072	0.222	1.000					
Activities	0.587 *	0.192	0.440	−0.428	0.355	0.315	1.000				
Independence	−0.042	−0.145	0.768 **	0.000	0.096	−0.086	0.435	1.000			
Relationship	0.248	−0.018	−0.141	0.065	0.808 **	0.183	0.460	0.144	1.000		
Emotions	0.391	0.334	0.062	−0.132	0.659 *	0.325	0.657*	0.290	0.756 **	1.000	
Body Image	0.102	−0.181	0.203	0.086	0.476	−0.104	0.475	0.424	0.573 *	0.608 *	1.000

*Note:* n = participant number; *SD* = standard deviation; *U* = *U* Mann–Whitney; *Z* = z punctuations; *p* = level of probability; * *p* ≤ 0.05; ** *p* ≤ 0.001; *r* = effect size.

## Data Availability

The data presented in this study are available on request by the corresponding author.

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
