# Peer review of "Health-Related Quality-of-Life Outcomes in Patients with Recessive and Dominant LGMD: A Comparative Cross-Sectional Study"

_muscles, 2025, doi:10.3390/muscles4030025_

Round 1
Reviewer 1 Report
Comments and Suggestions for Authors
Good morning,
First of all, I would like to congratulate you on the research carried out and thank you, as it is always enriching to learn from each researcher who develops a manuscript—especially when it concerns rare or low-prevalence diseases.
Secondly, I would like to share the following suggestions in case you find them useful to improve the publication:
-
In the methodology section, when discussing the population and sample size, you present a series of results describing whether there are significant differences between samples, etc. This content should be moved to the beginning of the results section, where the sample itself is analyzed in terms of sociodemographic factors or other specific characteristics.
-
I miss a section detailing the inclusion and exclusion criteria for the sample.
-
Lines 106 to 110: It would be helpful to include some kind of iconography or illustration to assist the reader, as I personally found this part difficult to follow.
-
I could not clearly identify the conclusions of the article. Perhaps they could be placed in a separate section outside the discussion, along with distinct sections for limitations and future directions.
Once again, thank you for the opportunity to review your article.
Best regards.
Author Response
Comments 1: In the methodology section, when discussing the population and sample size, you present a series of results describing whether there are significant differences between samples, etc. This content should be moved to the beginning of the results section, where the sample itself is analyzed in terms of sociodemographic factors or other specific characteristics.
Response 1: Thank you for pointing this out. We agree with this comment. Therefore, we have added this information in the Result section (Sociodemographic and clinical data of the sample) [page number: 4, paragraph: 4, line: 142-147]. This information has been included in red font in the manuscript.
Comments 2: I miss a section detailing the inclusion and exclusion criteria for the sample.
Response 2: Thank you for pointing this out. Information on inclusion and exclusion criteria was already included in the participants section. This information has been highlighted in red font for verification. [page number: 2, paragraph: 7, line: 86-89].
Comments 3: Lines 106 to 110: It would be helpful to include some kind of iconography or illustration to assist the reader, as I personally found this part difficult to follow.
Response 3: Thank you for pointing this out. We agree with this comment. Therefore, we have included a table (Table 1) to facilitate their understanding. [page number: 3, paragraph: 1, line: 110-111]. This information has been included in red font in the manuscript.
Comments 4: I could not clearly identify the conclusions of the article. Perhaps they could be placed in a separate section outside the discussion, along with distinct sections for limitations and future directions.
Response 4: Thank you for pointing this out. We agree with this comment. Therefore, we have separated the discussion into different sections, including ‘Limitations and future directions’ [page number: 9, paragraph: 3, line: 264] and ‘Conclusions’ [page number: 9, paragraph: 5, line: 288]. This information has been included in red font in the manuscript.

Reviewer 2 Report
Comments and Suggestions for Authors
The article proposed by Lépée-Aragón et al. addresses a relevant and underexplored topic: health-related quality of life (HRQoL) in recessive and dominant forms of limb-girdle muscular dystrophy (LGMD), using the INQoL instrument. The study is well-structured and largely adheres to the standards of empirical research (including ethical approval, informed consent, and appropriate statistical analysis). However, several aspects could be improved in terms of clarity, coherence, and scientific rigor:
- Please indicate the potential impact of the study on medical practice or the development of health policies.
- Please mention the criteria on which INQoL was chosen for this study and why it is superior or preferable to other instruments.
- The statistical result is expressed in a single sentence in the abstract, " which can be understood as a limitation of their social role due to the progression of the disease". I believe it should also be developed in the text.
Author Response
Comments 1: Please indicate the potential impact of the study on medical practice or the development of health policies.
Response 1: Thank you for pointing this out. This information had already been included in the discussion [page: 9; paragraph: 2]. However, in accordance with the review's recommendation, we have added information to clarify the manuscript's main contributions to clinical management and the development of health policies for these patients. This information has been included in red font in the manuscript. [page number: 2, paragraph: last one, line: 266-267].
Comments 2: Please mention the criteria on which INQoL was chosen for this study and why it is superior or preferable to other instruments.
Response 2: Thank you for pointing this out. To this respect, we have added a comment to clarify this point. This information has been included in red font in the manuscript.
[page number: 2, paragraph: last one, line: 95-96].
Comments 3: The statistical result is expressed in a single sentence in the abstract, "which can be understood as a limitation of their social role due to the progression of the disease". I believe it should also be developed in the text.
Response 3: Thank you for pointing this out. We agree with this comment. Therefore, a sentence has been included in the discussion section to provide further insight into this aspect. This information has been included in red font in the manuscript. [page number: 8, paragraph: 4, line: 214-216].
Reviewer 3 Report
Comments and Suggestions for Authors
This is a prospective, multi-center cross-sectional study evaluating the physical and psychosocial aspects of health-related quality of life in adult patients with different types of LGMD. It showed that recessive LGMD patients had a worse HRQoL compared to the dominant LGMD patients. Additionally, age was found to be associated with the relationship component on the questionnaire for the dominant LGMD patients, but not the recessive patients. Given scarce data in literature regarding the psychosocial outcomes in the LGMD patient population, this paper focuses on a novel and important question regarding the non-physical needs and challenges of these patients.
Additional comments:
- Although I agree with the choice of statistical methods, it should be clearly stated that with such small sample size and imbalance between the two groups (recessive vs. dominant LGMD), statistically significant findings should be interpreted with caution. This should be added as a limitation.
- What is meant by “the present cross-sectional research may shed some light on the different cases reported in the study by Angelini and Rodriguez”? This needs further clarification.
- Line 89: remove “have”
Author Response
Comments 1: Although I agree with the choice of statistical methods, it should be clearly stated that with such small sample size and imbalance between the two groups (recessive vs. dominant LGMD), statistically significant findings should be interpreted with caution. This should be added as a limitation.
Response 1: Thank you for pointing this out. We agree with this comment. Therefore, we have added this information in the section of “Limitations and future directions”. [page number:9, paragraph: 3, line: 268-269]. This information has been included in red font in the manuscript.
Comments 2: What is meant by “the present cross-sectional research may shed some light on the different cases reported in the study by Angelini and Rodriguez”? This needs further clarification.
Response 2: Thank you for pointing this out. We agree with this comment. By way of clarification, we would like to point out that the previous study by Angelini and Rodriguez was conducted only with case studies, and in the present study we have added a comparative research with a sample of patients divided into two groups. Therefore, we have tried to provide methodological and empirical support to the previous study. Moreover, we have highlighted this information in red font in the manuscript. [page number: 8, paragraph: 4, line: 214].
Comments 3: Line 89: remove “have”
Response 3: Thank you for pointing this out. We agree with this comment. Therefore, we have removed it. [page number: 2, paragraph: 7, line: 86].

Round 2
Reviewer 1 Report
Comments and Suggestions for Authors
Dear author,
thank you for taking care of mi considerations.
regards
Reviewer 2 Report
Comments and Suggestions for Authors
The authors have revised the manuscript and responded to previously requested suggestions. The manuscript has been substantially improved as a result and, in my opinion, now meets the standards required for publication. I therefore recommend its acceptance.